# Risk of Herpes Zoster in Patients with Pulmonary Tuberculosis—A Population-Based Cohort Study

**DOI:** 10.3390/ijerph20032656

**Published:** 2023-02-01

**Authors:** Chih-An Wang, Chia-Hung Chen, Wen-Che Hsieh, Tzu-Ju Hsu, Chung-Y. Hsu, Yung-Chi Cheng, Chao-Yu Hsu

**Affiliations:** 1Division of Respiratory Therapy, Ditmanson Medical Foundation, Chia-Yi Christian Hospital, Chia-Yi 600, Taiwan; 2Department of Medical Education, Ditmanson Medical Foundation, Chia-Yi Christian Hospital, Chia-Yi 600, Taiwan; 3Department of Medical Imaging, Ditmanson Medical Foundation, Chia-Yi Christian Hospital, Chia-Yi 600, Taiwan; 4Department of Chinese Medicine, Ditmanson Medical Foundation, Chia-Yi Christian Hospital, Chia-Yi 600, Taiwan; 5Management Office for Health Data, Clinical Trial Research Center, China Medical University Hospital, Taichung 404, Taiwan; 6Graduate Institute of Biomedical Sciences, China Medical University, Taichung 404, Taiwan; 7Department of Rehabilitation, Ditmanson Medical Foundation Chia-Yi Christian Hospital, Chia-Yi 600, Taiwan; 8Department of Artificial Intelligence and Healthcare Management, Central Taiwan University of Science and Technology, Taichung 406, Taiwan; 9Department of Medical Imaging and Radiological Sciences, Central Taiwan University of Science and Technology, Taichung 406, Taiwan; 10Center for General Education, National Taichung University of Science and Technology, Taichung 404, Taiwan; 11Department of General Education, National Chin-Yi University of Technology, Taichung 411, Taiwan

**Keywords:** tuberculosis, herpes zoster, population-based

## Abstract

Background: Pulmonary tuberculosis (TB), a global health problem, is typically caused by the bacterium *Mycobacterium tuberculosis*. Herpes zoster (HZ) is caused by the reactivation of the varicella-zoster virus (VZV). The reactivation of VZV can be caused by stress. We investigated whether pulmonary TB increases the risk of HZ development. Methods: This study used data that sampled a population of 2 million people in 2000 from the National Health Insurance Research Database. This cohort study observed Taiwanese patients aged 20–100 years with pulmonary TB from 2000 to 2017 (tracked to 2018). Pulmonary TB was defined as having two or more outpatient diagnoses or at least one admission record. To address potential bias caused by confounding factors, the control cohort and pulmonary TB cohort were matched 1:1 by age, gender, index year, and comorbidities. Patients with HZ before the index date were excluded. Results: A total of 30,805 patients were in the pulmonary TB and control cohorts. The incidence rate of HZ in pulmonary TB and control cohorts were 12.00 and 9.66 per 1000 person-years, respectively. The risk of HZ in the pulmonary TB cohort (adjusted hazard ratios = 1.23; 95% confidence interval = 1.16–1.30) was significantly higher than that of in control cohort. Among patients without comorbidities, the patients with TB were 1.28-fold more likely to have HZ than those without TB. Conclusion: Patients with TB should be well treated to avoid the potential risk of HZ occurrence. Although we identified the association between pulmonary TB and HZ, further studies are needed to confirm the result.

## 1. Introduction

Tuberculosis (TB) is a disease caused by *Mycobacterium tuberculosis*, which typically invades the lungs. According to a recent study, the global latent TB infection rate decreased from 30.66% in 1990 to 23.67% in 2019, an average annual percentage decrease of −0.9% [1]. In China, the incidence of TB was 66.61 per 100,000 person-years. Among these patients, almost 70% were men, and most of them were farmers and herdsmen [2]. In Taiwan, a high rate of latent TB infection in older patients in long-term care facilities was observed by using interferon-gamma release assays. The positive rate was 31.4% [3]. A meta-analysis showed that the prevalence of TB infection in health workers was 28% because they are in constant contact with patients [4]. During the COVID-19 pandemic, a meta-analysis study revealed that the combination of TB and COVID-19 infection was associated with a higher risk of fatality and an overall fatality rate of 13.9% from co-infection [5]. Although the prevalence of TB infection has declined over the last 30 years, the World Health Organization’s goal of eliminating TB has not yet been achieved [1]. TB remains a problem that cannot be ignored.

Varicella-zoster virus (VZV) commonly stays in the sensory nerve ganglia after primary infection. Herpes zoster (HZ) occurs with the reactivation of the VZV and is characterized by the appearance of painful herpetic blisters on the erythematous base. A meta-regression study found that the incidence of HZ increased with age, with 5.15 and 11.27 per 1000 person-years in patients aged 50–54 and >85 years, respectively. In addition, the incidence of HZ was higher in women than in men [6]. The reactivation of VZV can be caused by stress [7]. Stress from musculoskeletal-related [8,9,10,11] and urogenital-related [12,13,14] chronic pain and endocrine-related diseases [15] have been identified as being associated with HZ development. Pulmonary TB may be a stressor for infected individuals and may lead to the development of HZ. Therefore, we investigated whether pulmonary TB increases the risk of HZ development.

## 2. Materials and Methods

### 2.1. Data Source

The National Health Insurance (NHI) Research Database (NHIRD), a national data of Taiwan, contains information about almost all Taiwanese residents. This study used data, which covered Taiwan’s population of 1,998,311 people in 2000, obtained from the NHIRD. Disease diagnoses were coded according to the International Classification of Diseases, 9th Revision and 10th Revision, Clinical Modification (ICD-9-CM and ICD-10-CM).

### 2.2. Study Population, Outcome, and Comorbidities

This cohort study observed Taiwanese patients aged 20–100 years with pulmonary TB from 2000 to 2017 (tracked to 2018). Pulmonary TB (ICD-9-CM: 011; ICD-10-CM: A15.0, A15.4-A15.9) was indicated by the presence of two or more outpatient diagnoses or at least one admission record. To address control for confounders, a control cohort and pulmonary TB cohort were matched 1:1 by age, sex, index year, and comorbidities. Patients with HZ (ICD-9-CM: 053; ICD-10-CM: B02) before the index date were excluded. To reduce the effect of data selection bias, comorbidity is one of the confounding factors considered in this study. In the present study, the comorbidities were considered as diabetes mellitus (DM) (ICD-9-CM: 250; ICD-10-CM: E08-E13), chronic kidney disease (CKD) (ICD-9-CM: 585; ICD-10-CM: N18.4-N18.9), coronary artery disease (CAD) (ICD-9-CM: 410–414; ICD-10-CM: I20-I25) and cancer (ICD-9-CM: 140–208; ICD-10-CM: C).

### 2.3. Statistical Analysis

In the analysis, the variables of age, sex, and comorbidities are presented in terms of frequency and percentage. The variable of age is presented in terms of mean and ± standard deviation. A chi-square test and *t*-test were used to analyze the differences between categorical variables and continuous variables between the two groups. The incidence rate (IR) was calculated by using a logistic regression analysis. The multivariate Cox proportional hazards models with adjustment for sex, age, and comorbidities were used to estimate the risk of HZ in the patients with and without pulmonary TB. The cumulative incidence curve of pulmonary TB was obtained by using the Kaplan-Meier method. The curve was fitted using the log-rank test. All statistical analyses were performed using SAS software, version 9.4 (SAS Institute Inc., Cary, NC, USA). A graph of the cumulative incidence of pulmonary TB was obtained using R software. A two-sided *p*-value less than 0.05 indicated significance.

## 3. Results

The demographic factors, comorbidities, and follow-up time of the patients are shown in Table 1. Figure 1 shows the flowchart depicting the enrollment of patients from the NHIRD. A total of 30,805 patients were in the pulmonary TB and control cohorts. After propensity score matching, no significant differences in sex, age, comorbidities, and follow-up time were found. The mean age in the pulmonary TB cohort and control cohort was 62.27 (±18.04) and 62.21 (±18.02) years, respectively. The mean follow-up time in the pulmonary TB cohort and control cohort was 7.51 (±5.82) and 8.23 (±5.56) years, respectively.

Table 2 presents the IR and adjusted hazard ratios (aHR) of HZ associated with and without pulmonary TB and covariates. The IR of HZ in pulmonary TB and control cohorts were 12.00 and 9.66 per 1000 person-years. The IR of HZ in men and women were 10.86 and 10.63 per 1000 person-years. The IR of HZ in age groups of 20–29, 30–39, 40–49, and ≥50 years were 3.57, 4.42, 7.82, and 13.68 per 1000 person-years, respectively. Subsequently, the risk of HZ in the pulmonary TB cohort [aHR = 1.23; 95% confidence interval (CI) = 1.16–1.30] was significantly higher than in the control cohort. The risk of HZ in men (aHR = 1.07; 95% CI = 1.01–1.14) was significantly higher than in women. The risk of HZ in the age group of 40–49 (aHR = 2.05; 95% CI = 1.72–2.45) and ≥50 (aHR = 4.51; 95% CI = 3.84–5.30) were significantly higher than in those aged 20–29 years. The risk of HZ in patients with comorbidities was significantly higher than in those without comorbidities. The comorbidities included DM (aHR = 1.29; 95% CI = 1.21–1.38), CKD (aHR = 2.03; 95% CI = 1.79–2.31), CAD (aHR = 1.58; 95% CI = 1.47–1.69) and cancer (aHR = 1.96; 95% CI = 1.74–2.21). Figure 2 shows that the cumulative incidence of HZ in patients with pulmonary TB was significantly higher than that in patients without pulmonary TB, and the *p*-value of the log-rank test was less than 0.001.

Table 3 shows the results of further stratification, where the TB cohort (*n* = 2775) is compared with the control cohort (*n* = 2450). In the pulmonary TB cohort, the risk of HZ in men (IR = 12.06 vs. 9.79; aHR = 1.21; 95% CI = 1.13–1.29), women (IR = 11.91 vs. 9.45; aHR = 1.26; 95% CI = 1.15–1.38), aged 20–29 years (IR = 4.30 vs. 2.85; aHR = 1.46; 95% CI = 1.06–2.01), aged 40–49 years (IR = 8.74 vs.6.95; aHR = 1.21; 95% CI = 1.03–1.42), aged ≥50 years (IR = 15.27 vs.12.26; aHR = 1.22; 95% CI = 1.14–1.29), without comorbidities (IR = 10.42 vs. 7.95; aHR = 1.28; 95% CI = 1.19–1.38) and with comorbidities (IR = 15.22 vs. 12.88; aHR = 1.16; 95% CI = 1.07–1.26) were significantly higher than that of the control cohort.

## 4. Discussion

This is the first population-based study to identify the association between pulmonary TB and HZ. We found that the patients with pulmonary TB had a higher risk of HZ than those without pulmonary TB, with the crude hazard ratio (cHR) and aHR 1.20 and 1.23, respectively. In previous studies, the association between TB with several diseases, such as DM, CKD, cancer, and depression, has been identified. In addition, these diseases were correlated with the risk of HZ.

### 4.1. Tuberculosis-Diabetes Mellitus-Herpes Zoster

Wu et al. found a higher global prevalence and incidence of TB in patients with DM at 511.19/100,000 and 129.89/100,000 person-years, respectively, in a meta-analysis [16]. Compared with patients with glycated hemoglobin A1c (HbA1c) < 7.0%, those with HbA1c ≥ 7.0% had a 2.05-fold increased odds ratio for developing TB [17]. Patients with poorly controlled DM may have an increased risk of TB development. The authors suggested that TB screening in patients with uncontrolled diabetes is needed [17]. To estimate the incidence of DM among patients previously treated for TB, Salindri et al. found that in patients with no prior history of DM, the incidence of DM was 3.85 per 1000 person-years [18]. Lin et al. reported that the prevalence of DM in patients with TB was 13.73%. Among these patients, 19.32% were in the Americas, 17.31% were in Europe, 14.62% were in Southeast Asia, 13.59% were in the western Pacific, 9.61% were in the eastern Mediterranean, and 9.30% were in Africa [19]. The authors reported that combined TB and DM are prevalent worldwide [19].

Patients with DM have a significantly higher risk of developing HZ. In a meta-analysis of cohort studies, Lai et al. found that the incidences of HZ in patients with DM and without DM were 7.22 and 4.12 per 1000 person-years, respectively, with an IR of 1.60 [20]. In addition, Huang et al. reported that DM patients were at a higher risk than non-DM patients to have HZ, and the relative risk was 1.38 [21]. As the combination of TB and DM is common [19], the risk of HZ in the patient with TB or DM should not be ignored.

### 4.2. Tuberculosis-Chronic Kidney Disease-Herpes Zoster

A meta-analysis showed that the global incidence of TB in patients with CKD was estimated to be 3718 per 100,000, especially among those receiving hemodialysis (5611/100,000) [22]. A previous nationwide population-based study in Taiwan compared the risk of TB among patients with and without end-stage renal disease (ESRD). Hu et al. found that the patients with ESRD had a significantly higher risk of developing TB within 1 year and 1–2 years after diagnosis of ESRD, with IR of 4.13 and 2.12 compared with controls [23]. Shen et al. determined the CKD risk in patients with TB using Taiwanese NHI data with 8735 newly diagnosed patients with TB. The authors found that the incidence of CKD in the TB group was 1.27 times higher than that in the non-TB group [24]. That is, CKD was positively related to the risk of TB.

Li et al. reported that dialysis therapy was an independent risk factor for HZ in patients with CKD, with an odds ratio of 3.293. In addition, they also found that the patients with HZ had significantly lower total white blood cell and lymphocyte counts compared to the control group. Although that study only had 47 patients, their findings indicated the serious risk of HZ in patients with CKD [25]. HZ also has an influence on pre-dialysis patients with CKD. A nationwide population-based study reported that the IRs of HZ were 8.76 and 6.27 per 1000 person-years in patients with pre-dialysis CKD and patients without CKD, respectively. In addition, patients with CKD before dialysis had a 1.38-fold higher risk of developing HZ compared with patients without CKD [26]. Clearly, both CKD and pre-CKD are stressful for the involved individuals.

### 4.3. Tuberculosis-Cancer-Herpes Zoster

The global population-attributable fraction (PAF) of TB caused by cancers was 1.85%. The highest PAF was in Greenland (7.77%), followed by Canada (7.75%) and the United States of America (6.79%) [27]. A meta-analysis showed that the global incidence of cancer attributable to TB was 2.33%. The relative risks were 2.64 for head and neck cancer, 2.43 for hepatobiliary cancer, and 2.19 for Hodgkin’s lymphoma. Lung cancer ranked fourth with a relative risk of 1.69 [28]. Based on the finding of previous studies, TB is closely related to cancer.

Qian et al. found that adults with hematological and solid cancers had a higher risk of developing HZ than those without cancer, with relative risks of 3.74 and 1.30, respectively [29]. Similarly, Lin et al. found that the incidence of HZ in children with cancer (20.7/10,000 person-years) was higher than that in healthy children (2.4/10,000 person-years). The authors reported that all cancers increased the risk of HZ development, with leukemia having the strongest association (HR = 13.6) [30]. The aforementioned findings corroborate the relationship between TB, cancer, and HZ

### 4.4. Tuberculosis-Depression-Herpes Zoster

Not only physical diseases but also psychological problems are associated with TB. According to a meta-analysis, patients with TB have a relatively high prevalence of depression. Duko et al. reported a 45.19% prevalence of depression among patients with TB [31]. In addition, depression was strongly associated with negative TB treatment outcomes, with an odd ratio of 4.26 [32]. Thus, screening for depression is necessary for patients with TB [31], especially among women [33].

In patients with depression, the persistent activation of the hypothalamic-pituitary-adrenal axis may impair immune responses [34]. A weakened immune system may induce HZ reactivation. Two nationwide studies from the Korean and Taiwanese NHI databases demonstrated an increased risk of HZ in patients with depression with an HR of approximately 1.1 [35,36]. Stressors, whether of physical or psychological origins, may contribute to the development of HZ.

### 4.5. Comorbidities-Herpes Zoster

In this study, we found that the risk of HZ increased with the patients having DM, CKD, CAD, and cancer (Table 2). In Table 3, we found that among patients with comorbidities, the patients with TB have a higher risk of HZ (aHR = 1.16). However, among patients without comorbidities, TB patients still have a significantly higher risk of HZ occurrence. The patients with TB were 1.28-fold more likely to have HZ than those without TB. Therefore, TB might be a stressor that triggers HZ development.

### 4.6. Limitation

This retrospective study has several limitations. First, patients in this study were included based on the diagnosis codes of ICD-9. Disease severity may affect the immune system and lead to different outcomes. However, the severity of the disease cannot be distinguished due to the disadvantages of ICD-9. Second, smoking is considered to increase the risk of TB [37]. However, data on smoking were not included in the NHIRD. Third, TB should be treated for at least 6 months. The results may have been affected by patients who did not take their prescribed medication regularly. Nonetheless, most patients with TB in Taiwan receive proper treatment due to the implementation of NHI. Therefore, this limitation from medication noncompliance may have been slight. Fourth and finally, the HZ vaccine may influence the results of our study. However, the HZ vaccine was introduced to Taiwan in 2013. It is not covered by NHI, and only a few people receive the vaccine. Thus, this bias from HZ vaccination can also be ignored in this study.

TB is a global public health problem. This study, with nationwide and large sample size data, provides strong evidence and may become a reference for future research on population medicine.

## 5. Conclusions

Patients with TB should be well treated to avoid the potential risk of HZ occurrence. Although we identified the association between pulmonary TB and HZ, further studies are needed to confirm the result.

## Figures and Tables

**Figure 1 ijerph-20-02656-f001:**
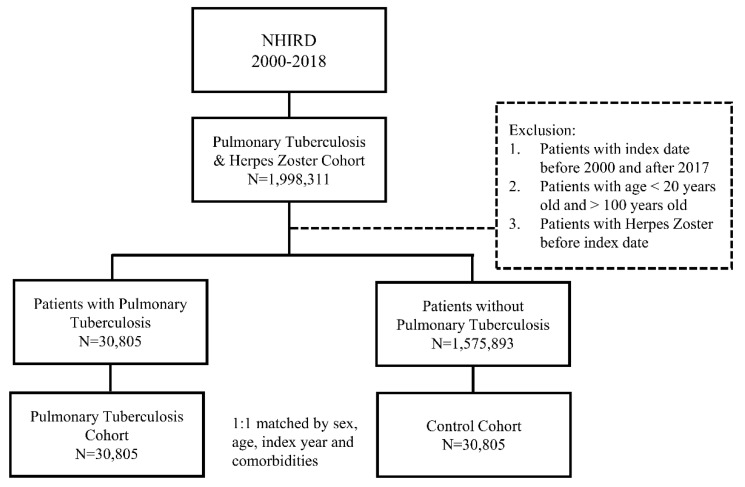
The flowchart of study sample selection from the National Health Insurance Research Database (NHIRD) of Taiwan.

**Figure 2 ijerph-20-02656-f002:**
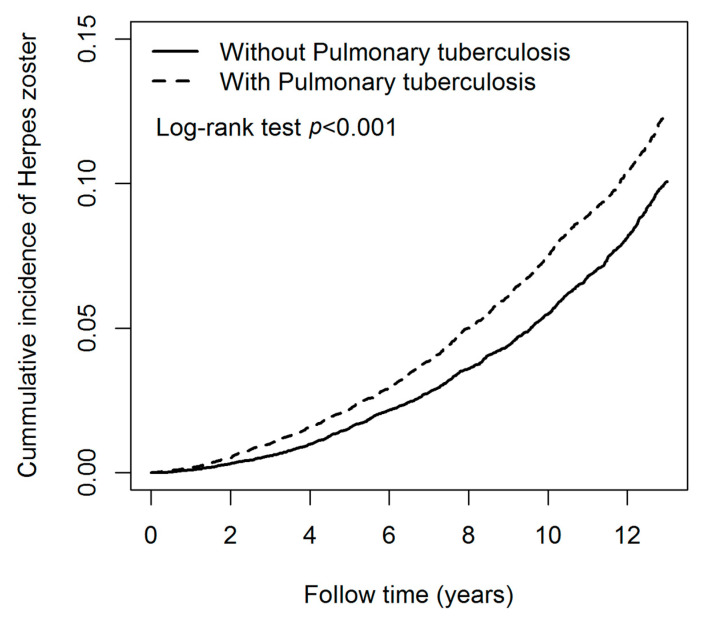
Kaplan-Meier curves of the cumulative incidence rate of herpes zoster during the follow-up period among patients with and without pulmonary tuberculosis.

**Table 1 ijerph-20-02656-t001:** Baseline characteristics for individuals with and without pulmonary tuberculosis.

Variables	Pulmonary Tuberculosis				*p*-Value
	No (N = 30,805)		Yes (N = 30,805)		
	*n*	%	*n*	%	
Gender					
Female	9753	31.66	9750	31.65	0.9793
Male	21,052	68.34	21,055	68.35	
Age, years					
20–29	1837	5.96	1835	5.96	0.9996
30–39	2339	7.59	2345	7.61	
40–49	3649	11.85	3653	11.86	
≥50	22,980	74.60	22,972	74.57	
mean ± SD ^a^	62.21	18.02	62.27	18.04	0.6732
Comorbidities					
DM	7813	25.36	7821	25.39	0.9410
CKD	1837	5.96	1861	6.04	0.6840
CAD	7543	24.49	7538	24.47	0.9626
Cancer	2063	6.70	2078	6.75	0.8093
Follow-up time, years					
mean ± SD ^a^	8.23	5.56	7.51	5.82	<0.001

^a^ *t*-test; SD: standard deviation; DM: diabetes mellitus; CKD: chronic kidney disease; CAD: coronary artery disease.

**Table 2 ijerph-20-02656-t002:** Incidences and hazard ratios of herpes zoster for individuals with and without pulmonary tuberculosis by gender, age, and comorbidities.

Variables	Herpes Zoster			cHR	(95% CI)	aHR	(95% CI)
	*n*	PY	IR				
Pulmonary tuberculosis							
No	2450	253,509	9.66	1.00	(reference)	1.00	(reference)
Yes	2775	231,214	12.00	1.20	(1.14, 1.27) ***	1.23	(1.16, 1.30) ***
Gender							
Female	1852	174,223	10.63	1.00	(reference)	1.00	(reference)
Male	3373	310,500	10.86	1.20	(1.13, 1.27) ***	1.07	(1.01, 1.14) *
Age, year							
20–29	157	43,997	3.57	1.00	(reference)	1.00	(reference)
30–39	238	53,801	4.42	1.22	(1.00, 1.49)	1.19	(0.97, 1.45)
40–49	616	78,810	7.82	2.24	(1.88, 2.67) ***	2.05	(1.72, 2.45) ***
≥50	4214	308,115	13.68	5.57	(4.75, 6.53) ***	4.51	(3.84, 5.30) ***
Comorbidities							
DM							
No	3960	388,726	10.19	1.00	(reference)	1.00	(reference)
Yes	1265	95,997	13.18	1.83	(1.72, 1.95) ***	1.29	(1.21, 1.38) ***
CKD							
No	4963	469,734	10.57	1.00	(reference)	1.00	(reference)
Yes	262	14,990	17.48	2.92	(2.58, 3.31) ***	2.03	(1.79, 2.31) ***
CAD							
No	3925	398,007	9.86	1.00	(reference)	1.00	(reference)
Yes	1300	86,716	14.99	2.40	(2.25, 2.56) ***	1.58	(1.47, 1.69) ***
Cancer							
No	4933	467,294	10.56	1.00	(reference)	1.00	(reference)
Yes	292	17,429	16.75	2.56	(2.27, 2.88) ***	1.96	(1.74, 2.21) ***

DM: diabetes mellitus; CKD: chronic kidney disease; CAD: coronary artery disease; PY: person-year; IR: incidence rate, per 1000 person-years; cHR: crude hazard ratio; aHR: adjusted hazard ratio, adjusted for age, sex, index year and comorbidities; CI: confidence interval; * *p* < 0.05, *** *p* < 0.001.

**Table 3 ijerph-20-02656-t003:** Cox proportional hazards regression analysis for the risk of herpes zoster.

Variables	Pulmonary Tuberculosis						cHR	(95% CI)	aHR	(95% CI)
	No			Yes						
	*n*	PY	IR	*n*	PY	IR				
Gender										
Female	854	90,409	9.45	998	83,814	11.91	1.22	(1.11, 1.34) ***	1.26	(1.15, 1.38) ***
Male	1596	163,100	9.79	1777	147,400	12.06	1.20	(1.12, 1.28) ***	1.21	(1.13, 1.29) ***
Age, year										
20–29	63	22,126	2.85	94	21,872	4.30	1.44	(1.05, 1.99) *	1.46	(1.06, 2.01) *
30–39	105	27,590	3.81	133	26,211	5.07	1.25	(0.96, 1.61)	1.28	(0.99, 1.65)
40–49	283	40,718	6.95	333	38,091	8.74	1.21	(1.03, 1.42) *	1.21	(1.03, 1.42) *
≥50	1999	163,075	12.26	2215	145,040	15.27	1.21	(1.14, 1.29) ***	1.22	(1.14, 1.29) ***
Comorbidities ^a^										
No	1314	165,321	7.95	1616	155,084	10.42	1.27	(1.18, 1.36) ***	1.28	(1.19, 1.38) ***
Yes	1136	88,188	12.88	1159	76,129	15.22	1.16	(1.07, 1.26) ***	1.16	(1.07, 1.26) ***

^a^ Individuals with any comorbidity of diabetes, chronic kidney disease, coronary artery disease, or cancer were classified into the comorbidity group; PY: person-year; IR: incidence rate, per 1000 person-years; cHR: crude hazard ratio; aHR: adjusted hazard ratio, adjusted for age, sex, index year and comorbidities; CI: confidence interval; * *p* < 0.05, *** *p* < 0.001.

## Data Availability

Data are available from the National Health Insurance Research Database, which is provided by the Taiwan National Health Insurance Administration. Due to the law of personal protection, data can be requested from Taiwan National Health Insurance Administration through a formal application (http://nhird.nhri.org.tw, accessed on 14 October 2022).

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
