# Peer review of "Risk of Herpes Zoster in Patients with Pulmonary Tuberculosis—A Population-Based Cohort Study"

_ijerph, 2023, doi:10.3390/ijerph20032656_

Round 1

Reviewer 1 Report

Risk of herpes zoster (HZ) in patients with pulmonary tuberculosis – population-based cohort study

Summary

Herpes zoster (shingle) is a clinical syndrome caused reactivation of latent varicella zoster virus (VZV). It occurs when the cell-mediated immune response is impaired und unable to maintain suppression of latent VZV reactivation.

This article studies a topic of scientific interest, given the still high infection rate of pulmonary tuberculosis in certain regions of the globe and the lack of specialized studies in the field of zoster epidemiology.

The main objective of the epidemiologic cohort study is to identify the likelihood that pulmonary tuberculosis (PTB) increases the risk of varicella-zoster virus reactivation.

One of the strengths of the present study is the large number of TBP patients of 30,805 selected from the 2 million people who represented the population of Taiwan included in the national health insurance database from 2000 to 2017 .

General concept comments

The observational cohort study was carried out on a large population of 1998311 people, from which 30,805 patients with PTB were selected. At the same time, the cohort of individuals without PTB, consisting of 30,805 patients, was selected.

I think that a more careful numerical characterization of the general population (2000000 people in the text or 1998311 in Figure 1) from which the patients and subjects of the control cohort were recruited is required. So, in „Material and Methods” (2.2. Study population, outcome and comorbidities), a clearer numerical description of all groups of patients is required.

Due to the impressive number of cases, the clinical trial will have great scientific value.

Specific comments

1.Introduction

The introduction of the article focuses on the infection rate and prevalence of tuberculosis in Asia, but also on the susceptible population.

In relation to the risk factors of VZV reactivation, these factors should be more clearly mentioned, especially those mentioned in lines 64-65.

It would be useful to specify the bibliographic source (if it exists) of the statement on lines 66-67.

Also, some theoretical information about the immune mechanisms participating in the reactivation of viral infection, especially in the context of bacillary infection, that is, mentioning the pathogenic mechanisms of the reactivation of VZV infection, would be useful for the readers.

It would be much better to clearly state the purpose of the study on lines 67-68.

2.Matherial and Methods

Patients included in the study were selected from the national health insurance database of Taiwan, NHIRD a reliable source with a total of 2 million patients (or 1998311 patients) who were followed for a long period of time (2000-2017).

The selection of patients was made on the basis of diagnostic codes, without reference to the clinical form of pulmonary tuberculosis. It would have been of great interest to identify which clinical forms of pulmonary tuberculosis increases the risk of varicella-zoster virus reactivation, but this objective can represent the subject of another article.

Also, the patients were divided into two groups (PTB patients and the control cohort), which consisted of a large number of patients (n=30,805) who were compared only in terms of age, sex and comorbidities, not including the other risk factors of viral reactivation (eg: emotional stress, previous use of corticosteroids, presence of malignancy, previous exposure to the virus).

3.Results and discussions

The most important aspect of the study is given the fact that it is the first observational, cohort study, which studies from an epidemiological point of view the association between the two clinical entities, as well as the fact that a correlation between the two conditions has been proven.

One of the important aspects of the study is represented by the confirmation of the association of the risk of developing herpes zoster with the comorbidities included in the study, comorbidities that through their evolution predispose to a decrease in cellular immunity, a decrease that is cited in the literature as one of the main reactivation factors of VZV infection. Also, the advanced age of the patients in the study was associated with the risk of viral reactivation, an aspect also mentioned in the literature.

 It would be useful for readers to make a correct numerical characterization of the randomized group of patients, in the text of the article, not only in Table 1.

That is the correct specification of the number of patients with TBP and HZ, respectively those with TBP without HZ.

Table 2 mentions the individuals from the cohort of those without PTB who developed HZ (2450 pts), as well as the individuals from the cohort of those with PTB who developed HZ (2775 pts). Does the term "event" refer to the number of individuals or to the number of events? It would be useful if the legend of the table were more explicit, clearly mentioning the number of patients with or without TBP who developed HZ.

In Table 3, it would be useful to mention that the cohort of those with TBP and HZ consists of 2775 patients so that the reader can make the connection much easier with the data mentioned in Table 2.

Clinical relevence

The clinical importance of the main objective is well supported, which is why a much more careful monitoring of patients with TBP and informing them about the possibility of HZ is required.

It is possible that this association represents the starting point of some researches, which will study the effects of the anti-zoosteria vaccine in populations with TBP.

The evolution of patients with PTB is of great interest, taking into account the immunosuppressive character of the disease, which in turn is a reactivation factor of the viral infection, although other risk factors were not included in the study and could represent another subject of research.

Author Response

Reviewer 1

The observational cohort study was carried out on a large population of 1998311 people, from which 30,805 patients with PTB were selected. At the same time, the cohort of individuals without PTB, consisting of 30,805 patients, was selected. I think that a more careful numerical characterization of the general population (2000000 people in the text or 1998311 in Figure 1) from which the patients and subjects of the control cohort were recruited is required. So, in „Material and Methods” (2.2. Study population, outcome and comorbidities), a clearer numerical description of all groups of patients is required.

Answer: Thanks for your comments We corrected the number of 1,998,311 in the text. We made the sentence “This study used data, which covered Taiwan’s population of 1,998,311 people in 2000, obtained from the NHIRD” in the text.

  1. Introduction

The introduction of the article focuses on the infection rate and prevalence of tuberculosis in Asia, but also on the susceptible population. In relation to the risk factors of VZV reactivation, these factors should be more clearly mentioned, especially those mentioned in lines 64-65. It would be useful to specify the bibliographic source (if it exists) of the statement on lines 66-67. Also, some theoretical information about the immune mechanisms participating in the reactivation of viral infection, especially in the context of bacillary infection, that is, mentioning the pathogenic mechanisms of the reactivation of VZV infection, would be useful for the readers. It would be much better to clearly state the purpose of the study on lines 67-68.

Answer: Thanks for your comments.

We added 2 sentences in the section of “Introduction” to make the text more clear. “Varicella-zoster virus (VZV) commonly stays in the sensory nerve ganglia after primary infection. Herpes zoster (HZ) occurs with the reactivation of the VZV, and is characterized by the appearance of painful herpetic blisters on the erythematous base.”

  1. Matherial and Methods

Patients included in the study were selected from the national health insurance database of Taiwan, NHIRD a reliable source with a total of 2 million patients (or 1998311 patients) who were followed for a long period of time (2000-2017). The selection of patients was made on the basis of diagnostic codes, without reference to the clinical form of pulmonary tuberculosis. It would have been of great interest to identify which clinical forms of pulmonary tuberculosis increases the risk of varicella-zoster virus reactivation, but this objective can represent the subject of another article.

Also, the patients were divided into two groups (PTB patients and the control cohort), which consisted of a large number of patients (n=30,805) who were compared only in terms of age, sex and comorbidities, not including the other risk factors of viral reactivation (eg: emotional stress, previous use of corticosteroids, presence of malignancy, previous exposure to the virus).

Answer: Thanks for your comments.

  1. Results and discussions

The most important aspect of the study is given the fact that it is the first observational, cohort study, which studies from an epidemiological point of view the association between the two clinical entities, as well as the fact that a correlation between the two conditions has been proven.

One of the important aspects of the study is represented by the confirmation of the association of the risk of developing herpes zoster with the comorbidities included in the study, comorbidities that through their evolution predispose to a decrease in cellular immunity, a decrease that is cited in the literature as one of the main reactivation factors of VZV infection. Also, the advanced age of the patients in the study was associated with the risk of viral reactivation, an aspect also mentioned in the literature.

 It would be useful for readers to make a correct numerical characterization of the randomized group of patients, in the text of the article, not only in Table 1.

That is the correct specification of the number of patients with TBP and HZ, respectively those with TBP without HZ.

Answer: Thanks for your comments.

Table 2 mentions the individuals from the cohort of those without PTB who developed HZ (2450 pts), as well as the individuals from the cohort of those with PTB who developed HZ (2775 pts). Does the term "event" refer to the number of individuals or to the number of events? It would be useful if the legend of the table were more explicit, clearly mentioning the number of patients with or without TBP who developed HZ.

In Table 3, it would be useful to mention that the cohort of those with TBP and HZ consists of 2775 patients so that the reader can make the connection much easier with the data mentioned in Table 2.

Answer: Thanks for your comments.

Sorry, we didn’t make it clear. The term "event" refers to the number of individuals. We change the word "event" to "n". It makes the table more clear. And, we added a statement on line 136. Thus, looks more link between Table2 and Table3.

Clinical relevence

The clinical importance of the main objective is well supported, which is why a much more careful monitoring of patients with TBP and informing them about the possibility of HZ is required. It is possible that this association represents the starting point of some researches, which will study the effects of the anti-zoosteria vaccine in populations with TBP. The evolution of patients with PTB is of great interest, taking into account the immunosuppressive character of the disease, which in turn is a reactivation factor of the viral infection, although other risk factors were not included in the study and could represent another subject of research.

Answer: Thanks for your comments

Reviewer 2 Report

It is a very interesting article.

The conclusion section should be improved. Also, english language editing are needed. 

Author Response

Reviewer 2

The conclusion section should be improved. Also, English language editing are needed.

Answer:

  1. Thanks for your comments.
  2. We change a sentence for the conclusion. “The patients with TB should be well treated and informed of the risk of HZ occurrence.”
  3. English language e diting of this paper has been edited.

Reviewer 3 Report

The study is an interesting and potentially impactful one with a large sample size that highlights the association between TB and HZ, with and without other comorbidities involved. It may help initiate screening and recognition for HZ in patients with TB. 

It seems as though the goal of the first paragraph of the introduction is to highlight that although the WHO aims to eliminate TB worldwide, it remains prevalent with especially high numbers in certain populations in China. It would be best to incorporate those statistics that highlight this point. Then in the second paragraph, once the authors have explained that stress may activate HSV, there should more discussion about how TB may act as a form of stress that triggers HSV. There should be a mention of whether any other studies have identified any relationship between HSV and TB before. Line 149 mentions that the comorbid diseases included here have been associated with TB, this may be worth mentioning earlier in the introduction or in the methods to explain why they were the 4 disease groups included as covariates. 

Methods are clearly defined. The results tables are very cleanly laid out and easy to follow. The paragraphs correlating with the tables, especially tables 2 and 3 should be more concise and highlight themes or specific statistics so as not to be redundant. 

Discussion is nicely organized. Makes it clear that each comorbidity, including depression, has links with both TB and HZ. Then explains that even without any comorbidity, TB may be a trigger for HZ. 

Author Response

Reviewer 3

The study is an interesting and potentially impactful one with a large sample size that highlights the association between TB and HZ, with and without other comorbidities involved. It may help initiate screening and recognition for HZ in patients with TB.

Answer: Thanks for your comments.

It seems as though the goal of the first paragraph of the introduction is to highlight that although the WHO aims to eliminate TB worldwide, it remains prevalent with especially high numbers in certain populations in China. It would be best to incorporate those statistics that highlight this point.

Answer: Thanks for your comments.

The arrangement of the first paragraph is followed globe -> China ->Taiwan. To inspect the severity of tuberculosis from the viewpoint of epidemiology.

Then in the second paragraph, once the authors have explained that stress may activate HSV, there should more discussion about how TB may act as a form of stress that triggers HSV. There should be a mention of whether any other studies have identified any relationship between HSV and TB before.

Answer: Thanks for your comments.

We discussed TB stress in the section of “Discussion”. We arranged and wrote “4.1. Tuberculosis-Diabetes mellitus-Herpes zoster”, “4.2. Tuberculosis-Chronic kidney disease-Herpes zoster”, “Tuberculosis-Cancer-Herpes zoster”, “4.4. Tuberculosis-Depression-Herpes zoster”. It means TB is associated with many diseases, TB must be stressful for affecting individuals and inducing the reactivation of herpes zoster.

To our knowledge, there is no paper to investigate the correlation between tuberculosis and herpes zoster. So, we used Taiwan’s National Health Insurance Research Database to identify the association. In the first paragraph of “Discussion”, we wrote “This is the first population-based study to identify the association between pulmonary TB and herpes zoster.”

Line 149 mentions that the comorbid diseases included here have been associated with TB, this may be worth mentioning earlier in the introduction or in the methods to explain why they were the 4 disease groups included as covariates.

Answer:Thanks for your comments.
In the section of “Method”, we added a sentence to address “c
omorbidity To reduce the effect of data selection bias, comorbidit y is one of the confounding fac tors considered in this study. In the present study, the comorbidities were considered as diabetes mellitus (DM) (ICD 9 CM: 250; ICD 10 CM: E08 E13), chronic kidney disease (CKD) (ICD 9 CM: 585; ICD 10 CM: N18.4 N18.9), coronary artery d isease (CAD) (ICD 9 CM: 410 414; ICD 10 CM: I20 I25) and cancer (ICD 9 CM: 140 208; ICD 10 CM: C).
(Ref: Zhang HW, et al. Enhanced Risk of Osteoporotic Fracture in Patients with Sarcopenia: A National Po pulation Based Study in Taiwan. J Pers Med. 2022 May 13;12(5):791. doi: 10.3390/jpm12050791.

Methods are clearly defined. The results tables are very cleanly laid out and easy to follow. The paragraphs correlating with the tables, especially tables 2 and 3 should be more concise and highlight themes or specific statistics so as not to be redundant.

Answer: Thanks for your comments.
The term "event" refers to the number of individuals. We corrected the word "event" to "n". It makes the table clear. And, we added a statement on line 136. Thus, looks more link between Table2 and Table3. “Table 3 shows the results of further stratification, where TB cohort (n = 2775) is compared with control cohort (n = 2450).”

Discussion is nicely organized. Makes it clear that each comorbidity, including depression, has links with both TB and HZ. Then explains that even without any comorbidity, TB may be a trigger for HZ.

Answer: Thanks for your comments.
